# Low Airspeed Impacts on Tom Turkey Response to Moderate Heat Stress

Derek Uemura [1], Prafulla Regmi [2], Jesse Grimes [3], Lingjuan Wang-Li [4] and Sanjay Shah [4,*]

1   Nelson Irrigation, Dayton, WA 99328, USA; uemuraderek@gmail.com
2   Department of Poultry Science, University of Georgia, Athens, GA 30602, USA; pregmi@uga.edu
3   Prestage Department of Poultry Science, NC State University, Raleigh, NC 27695, USA; jesse_grimes@ncsu.edu
4   Department of Biological and Agricultural Engineering, NC State University, Raleigh, NC 27695, USA; lwang5@ncsu.edu
*   Correspondence: sbshah3@ncsu.edu; Tel.: +1-919-515-6753

**Abstract:** Heat stress is a concern for turkeys in naturally ventilated houses. Chamber and room studies were used to assess heat stress at moderate temperatures (<25 °C) and low airspeeds on grown tom turkeys. In the chamber study, four ventilation rates × two temperatures (thermal comfort and thermal stress, 11 °C above thermal comfort) were applied to 13- to 19-week birds. Very small differences in airspeeds among the four treatments masked subcutaneous, cloacal, and infrared (IR) temperature differences at both temperatures. In the room study, four ventilation rates (0.07 m³· min⁻¹·kg⁻¹ or 100%, 75%, 50%, and 30% or Control) were applied to 21-week toms housed at <23 °C. The Control treatment had significantly higher whole-body and head temperatures vs. the other treatments. Only 100% had higher weight gain vs. 50%; hematology was unaffected by treatment. Higher ventilation rates reduced heat stress due to lower room temperatures, not airspeed differences, which were very low. The low-cost IR camera detected a heat stress difference ≥ 0.8 °C, corresponding to wind chill of 0.8 °C due to an airspeed of 0.8 m·s⁻¹ vs. still air on the USDA broiler wind chill curve. Machine vision combined with IR thermography could alleviate real-time poultry heat stress.

**Keywords:** infrared camera; ventilation; subcutaneous; wind chill; weight gain; hematocrit; corticosterone

## 1. Introduction

The United States (US) is the largest turkey producer in the world, producing 210 million birds worth $7.10 billion in 2022 [1]. In the US, turkeys are produced for meat and are mostly raised in confinement. Light and heavy turkey hens are marketed at 14 and 18 weeks at average market weights of 6.6 and 8.7 kg, respectively, while tom turkeys are marketed at 20 weeks at 20.4 kg [2]. However, market weights and ages vary with the integrator and geographical location.

Nearly all US broiler houses have tunnel ventilation, but most turkey houses are naturally ventilated due to economic reasons (J. Austin Baker, 17 July 2023, personal communication). Some US turkey houses are sidewall-ventilated. Few poultry houses are tunnel-ventilated in Brazil [3], the second biggest turkey producer. In Europe, poultry houses mostly use sidewall ventilation [4].

In natural ventilation, inside vs. outside temperature and pressure differences induce ventilation, which is aided by mixing fans. Mechanical ventilation is provided by fans and can be tunnel or sidewall (or cross) ventilation [5]. In tunnel-ventilated houses, fresh air enters through large openings at one end and the stale air is exhausted by fans at the opposite end of the house. Large volumes of air traveling along the length of the house produce high airspeeds (≥3 m·s⁻¹), which cools the birds. Sidewall-ventilated houses have

fans on the sidewall pushing fresh air or pulling exhaust air into or from the house, with the opposite screened wall serving as an outlet or inlet. Average airspeeds in sidewall-ventilated houses are <1 m·s$^{-1}$ [4], much lower than tunnel-ventilated houses even with the same ventilation rate (m$^3$·min$^{-1}$·kg$^{-1}$). Lateral air movement across a much larger cross-sectional area results in a much lower airspeed vs. the tunnel house, where it moves longitudinally across a much smaller cross-sectional area.

In mechanically ventilated houses, as air temperature increases, the ventilation rate is increased to reduce bird heat stress. The resulting airspeed provides a wind chill that reduces heat stress. Here, wind chill is defined as the reduction in temperature experienced by the bird due to some known airspeed vs. still air but at the same air temperature. As air temperature approaches bird surface temperature, airspeed alone is inadequate and evaporative cooling is provided to mitigate heat stress. In tunnel houses, evaporative cooling is provided using cool cell pads and/or foggers (or misters), while only foggers (or misters) can be used in sidewall-ventilated or naturally ventilated houses. Aviagen recommends activating evaporative cooling when air temperature exceeds 29 °C but not below 27 °C when ambient relative humidity (RH) exceeds 80% [6] to reduce the risk of wet litter that can increase ammonia levels and footpad concerns. However, in naturally ventilated houses, due to very low airspeeds, heat stress cannot be mitigated until evaporative cooling is activated. Hence, heat stress could start at much lower air temperatures in naturally ventilated houses vs. mechanically ventilated houses.

Heat stress, even at moderate temperatures, can adversely affect turkey welfare and performance. Turkeys ranging in age from 42 d to 140 d consumed more feed at 18 °C than at 28 °C. Average feed intake decreased by 2.3% per °C as temperature increased from 18 to 28 °C [7]. In a 4-week study, when 4-week-old tom and hen turkeys were exposed to five temperatures (12, 18, 24, 28, 32 °C), feed intake decreased linearly with increasing temperature while weight gain peaked at 18 °C [8]. Turkey breast meat yield declined even at temperatures in the range of 25 to 30 °C [9]. Older turkeys are in the danger zone when temperature and RH exceed 29 °C and 50% RH, respectively [10]. Hence, there is a need to assess the impact of low airspeed and moderate temperature conditions (prior to activation of foggers) on tom turkeys raised in naturally ventilated houses.

Modern turkeys are more susceptible to heat stress than the older genetic lines because they grow faster (higher metabolism) and are fed higher energy diets [11]. A 20.4 kg tom turkey from 1992–1998 produced 170% more heat than a turkey of the same size from 1974–1977 [12]. In the US, because modern turkeys are marketed at heavier weights, e.g., 14.4 kg in 2022 [1] vs. 10.0 kg in 1992 [13], they possess lesser surface area per unit mass and deeper cores that reduce their abilities to lose heat (vs. older turkeys), further increasing heat stress. Aviagen, a major turkey breeder, recommends a target air temperature of 12.8 °C for tom (male) turkeys 15 weeks or older [6].

Hence, the overall objective of this study was to evaluate the impacts of low airspeeds on tom turkeys 13 to 20 weeks of age under moderate temperatures (<25 °C). The specific objectives are listed below.

1.   Evaluate the short-term impact of four low airspeeds on heat stress experienced by tom turkeys housed in chambers at two different temperatures.
2.   Evaluate the longer-term impact of four low airspeeds on heat stress and performance of tom turkeys housed in rooms.

In the chamber study, heat stress was directly assessed with cloacal [14] and implantable [15] temperature measurements as well as IR thermometry [16], whereas in the room study, temperature measurements were made using implantable sensors and IR thermography. In the room study, heat stress was also assessed based on weight gain and blood chemistry [17]. Infrared thermography is being used to monitor heat stress and diseases in many livestock species, including poultry [18]. Hence, it may be possible to measure rapid wind chill based on surface temperature reduction.

## 2. Materials and Methods

This study was conducted at North Carolina State University's Talley Turkey Education Unit (NCSU-TTEU) in Raleigh, NC, USA. Additional details are in Uemura [18].

### 2.1. Chamber Study

This study was used to evaluate short-term temperature–airflow rate impacts on tom turkey temperature and, hence, heat stress under moderate temperatures (<25 °C). Two temperatures and four airflow rates were applied to tom turkeys housed in chambers for 2 h every week from 13 to 19 weeks of age.

#### 2.1.1. Chamber Description

Four identical chambers were constructed side-by-side on a wheeled 1.22 m W × 2.44 m L plywood platform (Figure 1). Each chamber had a footprint of 0.56 m L × 0.69 m W × 1.22 m H to provide a mature tom turkey 0.37 m$^2$ of space [6]. To minimize heat transfer between chambers, the chamber sidewalls and ceiling were insulated. The chamber had a welded wire screen (5 cm × 10 cm) that facilitated observation of the bird. On the opposite side of the door, there was a metal transition that tapered into a square hole. A 12 VDC 0.1 m φ variable speed fan (Make: JMC Datech; Model: 1225-12HBA; 3.24 m$^3 \cdot$min$^{-1}$ in free air) controlled with a speed controller (Make: Onyehn DC motor speed controller; Input: 1.8–15 VDC) was mounted on the square hole. Hence, air entering through the door was exhausted by the fan.

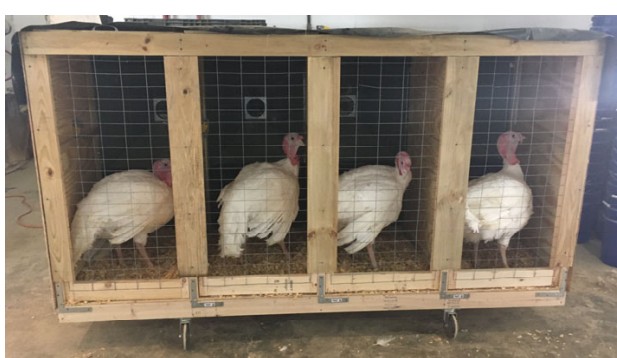

**Figure 1.** Chambers with turkeys. Note the fans at the back. The chambers were not instrumented at the time the image was taken.

Airflow rates for the various chambers are shown in Table 1. These airflow rates were obtained by changing the fan voltage, and the airflow rates were measured using a large vane anemometer (Make: Extech; Model: AN300; Range: 0.1–9999.9 m$^3 \cdot$min$^{-1}$; Accuracy: ± 1.5%) connected to the fan via a fitting plastic duct (to reduce turbulence). Airflow rates in chambers 1–3 were higher than the airflow rate of 1.3 m$^3 \cdot$min$^{-1} \cdot$bird$^{-1}$ (0.07 m$^3 \cdot$min$^{-1} \cdot$kg$^{-1}$ for 19.1 kg toms) in tunnel houses because the fan did not operate at <5 V. Despite such airflow rates, average airspeeds (Table 1) were very low due to the large inlet area. Given the larger door size, it was assumed that the measured airflow rate or calculated airspeeds would not be affected by the presence or absence of the bird in the chamber. Smoke tests were used to visually confirm that the fan was moving air through the chamber in the entire range of airflow rates measured.

Since the chamber study was conducted in an unheated room in winter, to ensure that all the chambers had the same temperature (regardless of ventilation rate), a 100 W ceramic heat lamp in an aluminum brooder lamp holder was suspended from the ceiling of the chamber. The lamp was controlled by its own thermostat (Make: Inkbird; Model: ITC-306T; Accuracy: ±0.1 °C; Range: −50 °C to 70 °C).

**Table 1.** Airflow rates and airspeeds assigned to the different treatments in the chamber study.

| Chamber No. (Treatment) | Airflow Rate, $m^3 \cdot min^{-1}$ | Average Airspeed [1], $m \cdot s^{-1}$ |
|---|---|---|
| 1 (High) | 2.9 | 0.07 |
| 2 (Medium) | 2.4 | 0.06 |
| 3 (Low) | 1.5 | 0.04 |
| 4 (Control) | 0 | 0 |

[1] Airflow rate divided by the vertical cross-sectional area (0.56 m × 1.22 m).

### 2.1.2. Turkey Management

Twenty 12-week-old tom turkeys (Large White variety, Nicholas Select strain) were distributed equally between two holding pens (2.44 m × 2.44 m ea.) with welded-wire sides in a naturally ventilated barn. The pens had fresh pine wood bedding, and grower feed and water were provided ad libitum. The turkeys were implanted with temperature transponders (Make: BMDS; Model: IPT-300 HTEC; Accuracy: ±0.2 °C from 37 to 42 °C) in the back between the two scapulae under the skin using the supplied sterile hypodermic needles. Each transponder was programmed and given an identification number with the handheld reader (Make: BMDS; Model: DAS-8027-IUS).

During the chamber study, the turkeys were acclimated in the chambers (with ~5 cm of litter) for 15 min with three 2.54 cm ϕ wooden dowels straddling their legs to keep them standing. This process was repeated for 5 d. Previous trials had shown that bird surface temperature varied based on whether the turkey was sitting or standing.

### 2.1.3. Treatment Design and Monitoring

The two temperatures used were either thermal comfort (TC) or thermal stress (TS). Optimum temperatures suggested by Aviagen [6], a poultry breeding company, for weeks 1 to 14 extrapolated to week 19, were used as TC (assumed to be the mid-point of the zone of thermal comfort) for weeks 13 to 19. The TS treatment was selected as 11.1 °C (20 °F) above TC for each respective week. TC decreased linearly from 13.3 °C to 10.2 °C, whereas TS decreased linearly from 24.4 °C to 21.3 °C (Figure 2).

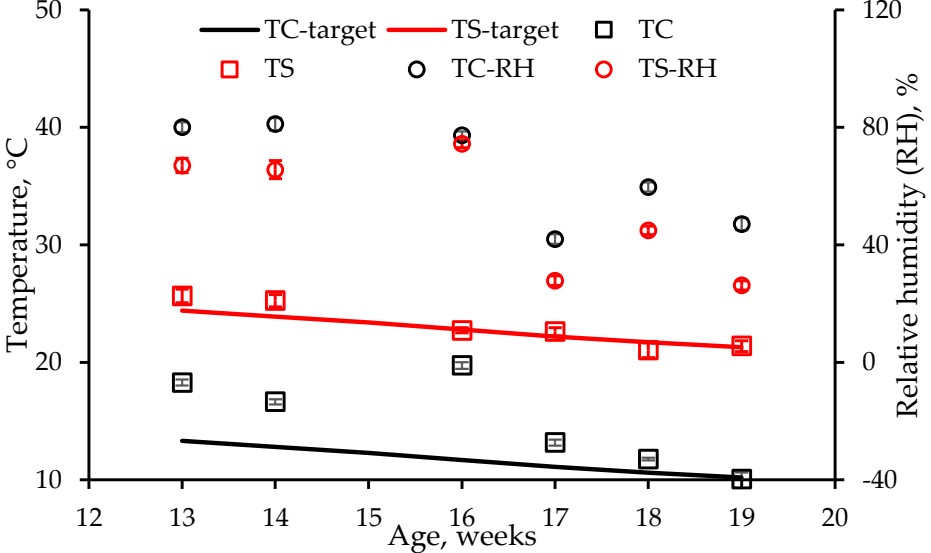

**Figure 2.** Target and measured thermal comfort (TC) and thermal stress (TS) temperatures in the chambers for weeks 13–19. Measured relative humidity (RH) values in the TC and TS treatments are also plotted. Measured temperature and RH data points are averages of the four chambers (treatments), and their standard deviations (SD) are also shown. Data for week 15 were excluded since surface temperature data were lost.

The chamber platform was placed in the corner of the room, and the space was screened on two sides using plastic sheeting. As needed, a portable air conditioner (A/C) was used to cool the enclosure as close to the optimum temperature (Figure 2); however, it was limited to a minimum cooling temperature of 15.6 °C. If the enclosure temperature was lower than the optimum temperature (Figure 2), two thermostatically controlled oscillating ceramic heaters were used to increase the enclosure temperature.

Every week, monitoring was performed on two consecutive days. On both days, in the morning (9–11 a.m.), all four chambers were operated as close as possible to the optimum or TC temperature, whereas in the afternoon, they were operated as close as possible to the TS temperature. Chambers 1 through 4 had different airflow rates, as shown in Table 1. Hence, for each airflow rate–temperature treatment, there were two replicates. Feed was removed from the holding pens 1 h prior to transfer of the turkeys into the chambers to minimize any effects from heat produced due to feed intake. A turkey was selected from one of two pens at random, weighed on a digital scale, and transferred to one of the chambers (with ~5 cm of litter) after its cloacal temperature had been measured with a digital thermometer (Make: Walgreens 30 Second Digital Thermometer; Model: WIC#991454; Accuracy: ±0.2 °C; Range: 32.0–43.9 °C; Resolution: 0.1 °C). A bird used in the morning was not used in the afternoon. As mentioned above, the turkey was kept standing using the wooden dowels for the entire 2 h of monitoring.

After placing the turkeys in the chambers, initial subcutaneous temperatures were measured using the handheld reader. In addition, IR and digital images were captured simultaneously using the Teledyne FLIR E8 IR camera; the camera's specifications are in Table 2. Prior to the start of the study, the IR camera had been calibrated in the lab [19]. The IR camera's emissivity ($\varepsilon$) was set to 0.95, ambient temperature to the target temperature, and distance to 0 m. An $\varepsilon$ of 0.95 was the most widely used value for poultry bird surfaces [20]. The IR camera was pointed perpendicular to the chamber to capture the entire turkey on the screen. Subcutaneous temperatures and IR images were captured every 30 min post-placement. At the end of the experiment, final cloacal temperatures were measured, and the turkeys were returned to their holding pens.

**Table 2.** Teledyne FLIR E8 IR camera used in the study.

| Parameters | Values |
| --- | --- |
| Accuracy | greater than ±2 °C or ±2% of reading [1] |
| Surface temperature range | −20 °C to 250 °C |
| Thermal sensitivity | <0.06 °C |
| Spectral range | 7.5 to 13 µm |
| IR resolution | 76,800 (320 × 240) pixels |
| Digital resolution | 307,200 (640 × 480) pixels |
| Field of view | 45° × 34° |

[1] Ambient temperature range of 10 °C to 35 °C when the surface temperature is >0 °C.

A data logger (Make: Tinytag Ultra; Model: TGU-1500; Accuracy: ±0.2 °C; Range: 0 to 70 °C) was placed in each chamber to measure and log temperature and relative humidity (RH) every 15 min during the confinement period. Since measured temperatures were similar among the four chambers, as indicated by low standard deviations (SD), they were averaged for each week (Figure 2). Average weekly measured and target TS temperatures were within ±2 °C. However, in the TC regime, higher measured temperatures in all weeks except week 19 (Figure 2) were due to the inability of the A/C unit used to cool the enclosure despite the study being conducted in the morning during the cool season. Relative humidity in both temperature regimes ranged from 60 to 85% during weeks 13 and 14 of age due to warm, humid ambient conditions. From weeks 17 through to 19 of age, RH ranged from 30 to 60% due to cooler and drier conditions (Figure 2).

### 2.1.4. Infrared Image Analysis

The FLIR Thermal Studio Pro (FLIR, Wilsonville, OR, USA) software (https://www.flir.com/support/products/thermal-studio-pro/#Overview) was used to analyze turkey IR images. Since this software allowed analysis of surface temperature on user-specified shapes, the whole body, as well as specific sections (head, torso, and legs), were analyzed for temperature. It may be noted that all IR temperature measurements (also in the Room Study) were adjusted based on calibration of the IR camera [19].

### 2.1.5. Statistical Analyses of Chamber Data

Since there were only two replicates per treatment, parametric statistics could not be used for hypothesis testing. Instead, the non-parametric Wilcoxon/Kruskal–Wallis test was used to compare treatment effects on surface, subcutaneous, and cloacal temperatures using JMP Pro 15 software (Cary, NC, USA). If there was a significant treatment effect, the means were compared using the Steel–Dwass method for all pairs. Throughout the study, $\alpha$ of 0.1 was used because there were only two replicates per treatment, and within-treatment variability was very high.

### 2.2. Room Study

This study was used to compare turkey heat stress and performance at four different ventilation rates at a target room temperature <20.5 °C during the winter. Hence, the target room temperature was 11 °C above TC in the chamber study, and thus at the TS level to thermally stress the birds slightly.

### 2.2.1. Room Description

This study was conducted at the TTEU in a building with sixteen climate-controlled rooms, with eight rooms on either side of a hallway. Each room (4.27 m L × 2.89 m W × 2.59 m H) was equipped with its own environmental controller, exhaust fan, timed lighting, propane heaters, feeders, and waterers. All the walls and the ceiling were insulated, and the floor was concrete. To reduce edge effects from the ambient temperature, four rooms in the center on the east side of the building were used. Each room was divided into two halves with a screen partition, and the fan-end of the room (2.13 m L × 2.89 m W) (Figure 3), was used to house 15 toms at 0.41 m²·tom⁻¹. A thick plastic curtain stapled to the ceiling was lowered to leave a 0.76 m opening above the litter. Fresh air entered through the air inlets at the top of the room attached to the plastic curtain and traveled down to enter the space occupied by the birds to mimic airflow in a turkey barn. The ventilation rate (discussed below), divided by the open area (2.20 m²) at the bottom of the curtain, was used to calculate the air velocity.

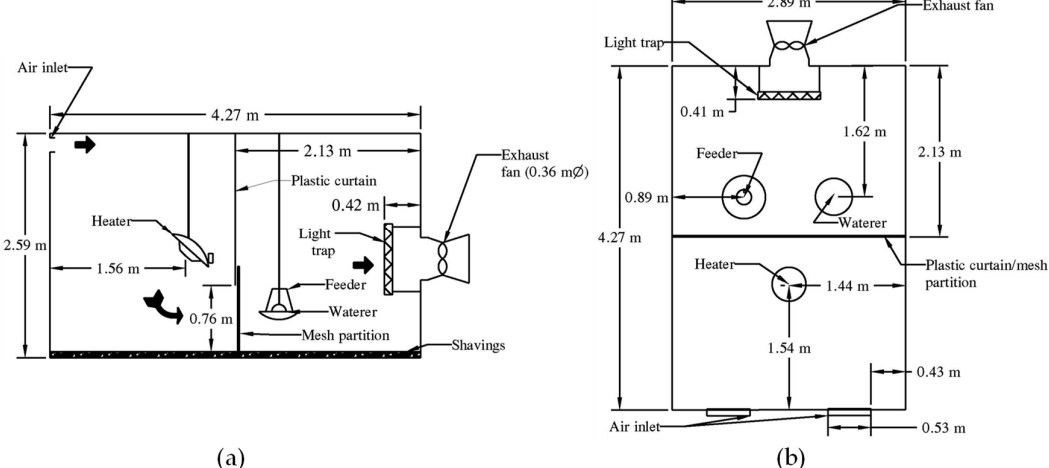

**Figure 3.** Room layout—(**a**) elevation and (**b**) plan. Drawn to scale.

### 2.2.2. Treatments

Sixty 20-week tom turkeys (Large White variety, Nicholas Select strain), which had previously been used in a heat stress study, were divided equally amongst the four rooms from 5 December through 11 December 2020. There were four treatments, with each treatment applied to a room based on the live weight of the 15 tom turkeys in that room. For this, six birds from each room were weighed 2 d prior to confinement, and the average bird weight was multiplied by 15 (number of birds per room) to calculate the total live weight; this total live weight was then used to calculate the ventilation rate for each treatment. After initial weighing, for the 2 d prior to the start of the study, all four rooms had the same ventilation rate. Beginning 5 December 2020, three of the rooms received ventilation rates of 100% (VR-100), 75% (VR-75), and 50% (VR-50) (Table 3) of the hot weather ventilation rate ($0.07 \text{ m}^3 \cdot \text{min}^{-1} \cdot \text{kg}^{-1}$) provided in commercial turkey houses [5]. The ventilation rate in the Control room was calculated by performing a sensible heat balance following [21] the use of heat production by tom turkeys reported in [12]. The sensible heat balance also accounted for heat losses and gains through the room envelope. The calculated ventilation rate of $4.5 \text{ m}^3 \cdot \text{min}^{-1}$ for the Control room had to be increased to $5.5 \text{ m}^3 \cdot \text{min}^{-1}$ 2 d after the start of the study due to excessive RH.

**Table 3.** Airflow rates and airspeeds in the different treatments in the room study.

| Room | Ventilation Rate, $\text{m}^3 \cdot \text{min}^{-1}$ | Calculated Average Horizontal Airspeed [1], $\text{m} \cdot \text{s}^{-1}$ |
|---|---|---|
| VR-100 | 17.0 | 0.13 |
| VR-75 | 13.6 | 0.10 |
| VR-50 | 9.1 | 0.07 |
| Control | 5.5 | 0.04 |

[1] Ventilation rate divided by area of opening (Figure 3).

The ventilation rate was set to the desired value by adjusting the fan speed on the environmental controller (Make: Munters Aerotech; Model: Aerospeed 1.1, Series 5) while measuring the airflow rate using a balometer (Make: Testo; Model: 420; Accuracy: $\pm$(3% of measured value + $0.2 \text{ m}^3 \cdot \text{min}^{-1}$)). Hardboard with a hole matching the diameter of the fan cone was placed over the fan cone, and the balometer was pressed against the opening to measure the volumetric flow rate. In each room, the fan was run continuously at constant speed, and the propane heater was set to maintain a room temperature of 20.5 °C.

### 2.2.3. Turkey Management and Monitoring

Blood was drawn from the six birds in each room that had been weighed prior to placement, both at the beginning and end of the 6 d study. These blood samples were stored in a refrigerator until they were analyzed for hematocrit and plasma corticosterone. Hence, in addition to comparison among the four treatments ($n = 6$), the initial and final blood parameters were statistically compared using analysis of variance (ANOVA) with means comparison being made with Tukey's Honest Significant Difference (HSD) using $\alpha = 0.1$; see Uemura [18] for details. Five of six birds in VR-100, VR-75, and Control treatments had received the high-temperature treatment (30 °C), whereas all six birds in the VR-50 treatment had been in the high-temperature treatment in the previous study.

In each room, the core body temperature of three out of six birds weighed was monitored with an implantable sensor (Make: BMEDiCAL; Model: Anipill; Accuracy: $\pm$0.2 °C) inserted between the scapulae, as described in [18]. After insertion, the turkey was returned to its room. Immediately before insertion, the Anipill was activated and assigned an identification number. The Anipill recorded temperatures every 15 min. In each pair of rooms, a data logger displayed and stored temperature data from the six birds (three per room) through wireless telemetry. The datalogger had an effective range of ~3 m.

The turkeys were placed on built-up litter top-dressed with pine shavings. The birds were wing-tagged and confined to half of the fan-end of the room (Figure 2), which afforded them $0.41 \text{ m}^2$ per bird. Feed and water were provided ad libitum (grower phase feed).

TinyTag data loggers (also used in the chamber study), suspended using twine 0.91 m above the litter, recorded room temperature and RH every 5 min. Bird surface temperatures were measured in each room once daily for 6 d using the FLIR IR camera. The IR images were analyzed using the method used in the chamber study. Bird core and surface temperatures (measured with the IR camera) were compared among the treatments using ANOVA, and Tukey's HSD was used to compare the treatment means at $\alpha = 0.1$.

Average weight gain was the average difference between the final and initial weights of the six birds in each room and compared among the treatments using ANOVA and Tukey's (HSD) at $\alpha = 0.1$. Since the birds were weighed 2 d before the start of the study, weight gain occurred over 8 d. Due to missing data, the feed conversion ratio was not calculated.

## 3. Results and Discussion

### 3.1. Chamber Study

Surface, subcutaneous, and cloacal temperatures of tom turkeys housed in chambers subjected to four airflow rates (Table 1) and two temperature treatments were measured. The two temperature treatments were changed weekly (Figure 2).

### 3.1.1. Cloacal vs. Subcutaneous Temperatures

Mean subcutaneous and cloacal temperatures, combined for all days and treatments, were $40.7 \pm 0.4$ °C and $40.9 \pm 0.3$ °C, respectively, and the difference was within the measurement accuracy ($\pm 0.2$ °C) of either sensor. The root mean square error (RMSE) for 112 pairs of data points was 0.2 °C, indicating low variability. The subcutaneous transponder had a positive bias of 0.2 °C. However, the subcutaneous and cloacal temperature readings were only moderately correlated ($r^2 = 0.58$), which indicated that the subcutaneous transponder responded more rapidly to environmental conditions. Hence, subcutaneous temperature could be an early indicator of the bird's heat stress.

### 3.1.2. Turkey Surface Temperatures

Airflow rate did not significantly ($p > 0.1$) affect turkey temperature (average cloacal, subcutaneous, body surface, or head surface) (Table 4) on any week or at either chamber temperature (TC or TS). The lack of treatment effect, despite large differences in airflow rate (Table 1), could have been due to the low average airspeeds that were also similar among the treatments. Other contributing factors could have been a low number of replicates and variability between birds.

**Table 4.** Statistical comparison of airflow rate effects (2.9, 2.4,1.5, and 0 m$^3$·min$^{-1}$, Table 1) on whole body, head, subcutaneous, and cloacal temperatures at TC or TS temperatures measured during weeks 13 to 19 of age, except week 15 when IR temperature data were lost.

| Week–Temperature | Wilcoxon/Kruskal–Wallis Test $p$ Value | | | |
|---|---|---|---|---|
| | **Whole Body** | **Head** | **Cloacal** | **Subcutaneous** |
| 13-TC [1] | 0.244 | 0.478 | 0.299 | 0.126 |
| 13-TS | 0.961 | 0.908 | 1.000 | 0.245 |
| 14-TC | 0.315 | 0.112 | 0.944 | 0.740 |
| 14-TS | 0.919 | 0.682 | 0.278 | 0.364 |
| 16-TC [2] | 0.392 | 0.392 | 0.414 | 0.530 |
| 16-TS | 0.419 | 0.212 | 0.747 | 0.761 |
| 17-TC | 0.418 | 0.881 | 0.865 | 0.145 |
| 17-TS | 0.682 | 0.380 | 0.685 | 0.803 |
| 18-TC | 0.983 | 0.761 | 0.566 | 0.988 |
| 18-TS | 0.982 | 0.919 | 0.747 | 0.321 |

**Table 4.** *Cont.*

| Week–Temperature | Wilcoxon/Kruskal–Wallis Test $p$ Value | | | |
|---|---|---|---|---|
| | Whole Body | Head | Cloacal | Subcutaneous |
| 19-TC [2] | 0.392 | 0.392 | 0.199 | 0.131 |
| 19-TS | 0.869 | 0.119 | 0.139 | 0.209 |

[1] First two numbers are weeks of age, and the following letters are either TC (thermal comfort) or TS (heat stress) temperature. [2] Based on single replicate.

Infrared images of 18-week toms, when measured chamber temperatures at both temperature regimes were close to the target temperatures, are shown in Figure 4. The IR image of a specific body section (e.g., head) was used to calculate the average temperature for that section. In the torso section, feathers not in contact with the body (e.g., TC-4, Figure 4) appeared cool and reduced average section temperature. The featherless head and leg sections were warmer than the torso. The TS birds appeared warmer than the TC birds.

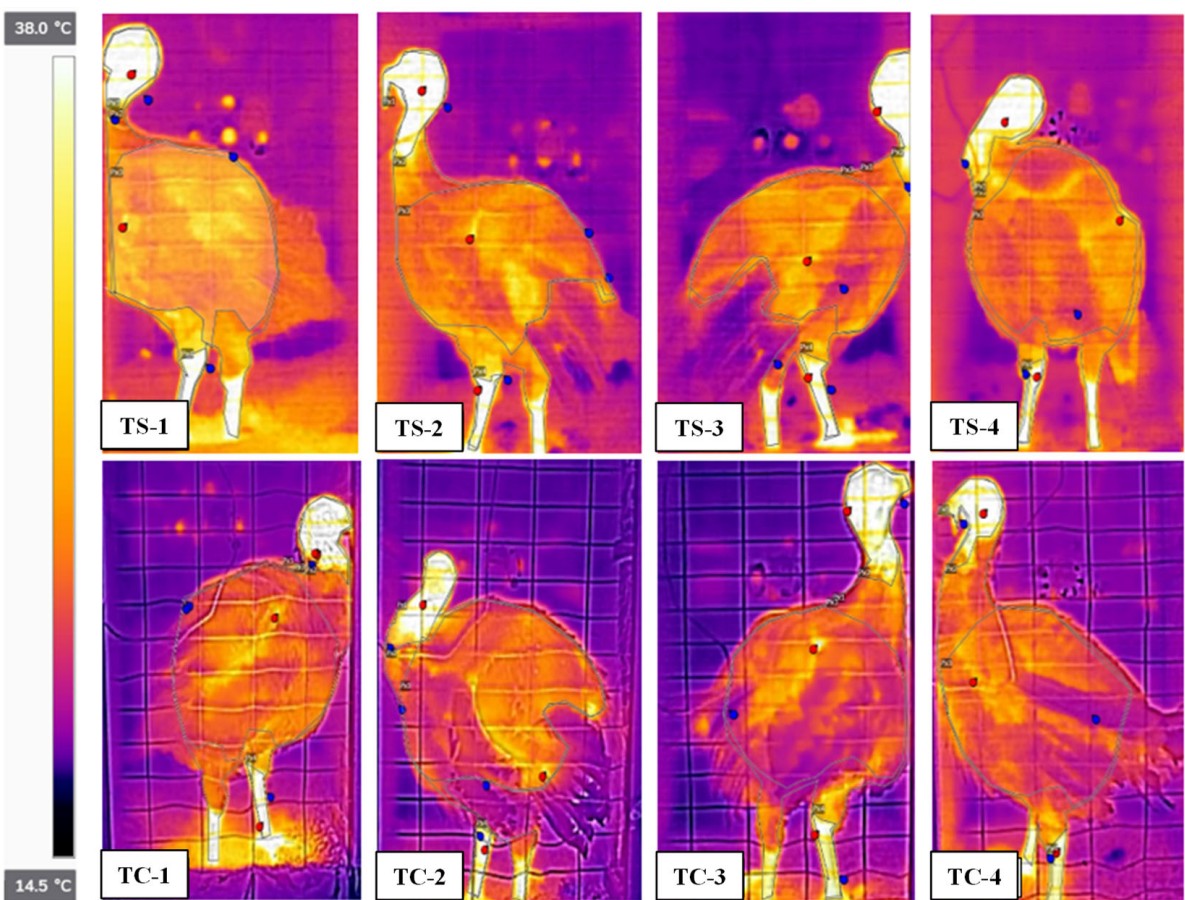

**Figure 4.** Infrared images of 18-week toms during 2 h of exposure in chambers at one of two target temperatures (TC or TS) and one of four airflow rates (1, 2, 3, or 4). All images have the same temperature scale. Red and black dots indicate the maximum and minimum temperatures of each section (head, torso, and legs), respectively. Temperatures are thermal comfort (TC) and thermal stress (TS = TC + 11.1 °C). Airflow rates are 2.9 m$^3$·min$^{-1}$ (High, 1), 2.4 m$^3$·min$^{-1}$ (Medium, 2), 1.5 m$^3$·min$^{-1}$ (Low, 3), and 0 m$^3$·min$^{-1}$ (Control, 4).

Mean change ($n$ = 2) in the cloacal, subcutaneous, average head, and average torso temperatures for the 18-week toms over 2 h of exposure in the chamber are compared in Figure 5. Data for other weeks are in [18]. Temperature rise had been expected to decrease (or even reverse) with increased airflow rate at the same chamber temperature. However,

this trend was only fully evident for subcutaneous temperature in the TS temperature regime and, to a lesser extent, for subcutaneous temperature in the TC regime and cloacal temperatures in both TC and TS regimes (Figure 5a). As expected, the subcutaneous temperature increase was generally higher in TS (vs. TC) and inversely correlated with airflow rate. Head and torso surface temperatures seemed to be unaffected by either airflow rate or chamber temperature (TC vs. TS) (Figure 5b). In chamber studies, subcutaneous temperature appears to be more reliable for monitoring short-term heat stress.

When studying multiple birds, subcutaneous temperature monitoring may not be feasible, and due to its non-invasive nature, the IR camera would be preferable. However, the use of IR images for quantitative determination was challenging for several reasons. Infrared surface temperatures were affected by protruding feathers (as mentioned earlier, Figure 4) and bird movement (especially the head). There was uncertainty in delineating the section of interest to determine the average surface temperature. Differences in the orientation of the birds with respect to the IR camera increased the uncertainty. Finally, the differences in airspeeds between the treatments (Table 1) were too low to cause differences in surface temperature that could be detected by the IR camera.

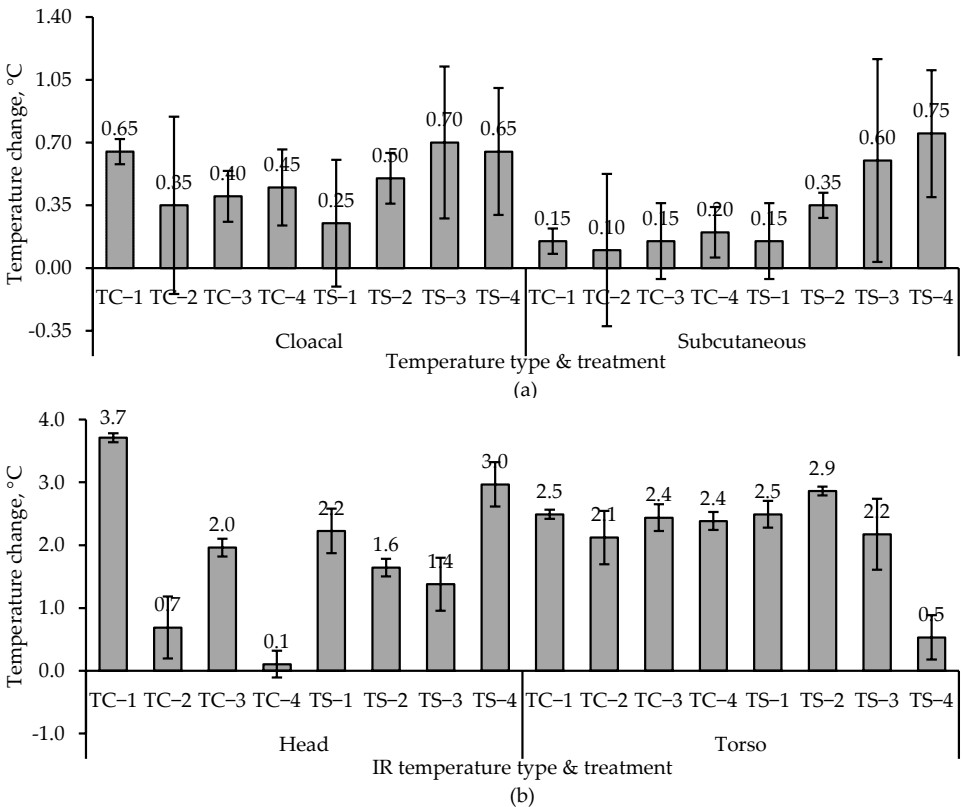

**Figure 5.** Comparison of the mean (*n* = 2) temperature change in 18-week tom turkeys during 2 h of exposure in chambers with one of two (TC or TS) temperatures and one of four airflow rates (1, 2, 3, or 4). (**a**) Average cloacal and subcutaneous temperatures and (**b**) average head surface and torso surface IR temperatures. The error bars indicate standard deviation. Temperatures are thermal comfort (TC) and thermal stress (TS = TC + 11.1 °C). Airflow rates are 2.9 $m^3 \cdot min^{-1}$ (High, 1), 2.4 $m^3 \cdot min^{-1}$ (Medium, 2), 1.5 $m^3 \cdot min^{-1}$ (Low, 3), and 0 $m^3 \cdot min^{-1}$ (Control, 4). The two figures have different *y*-axis scales.

Since there were no treatment effects on any temperature measurement (Table 4), temperatures for all four chambers for TC or TS were averaged (Figure 6). Generally, the TS cloacal and subcutaneous temperatures were higher than the corresponding TC values (Figure 6). However, temperature differences between TC and TS fluctuated due to difficulty in controlling chamber temperatures, particularly in the TC treatment, as well as

fluctuations in RH (Figure 2). Whereas weekly TS head temperatures were slightly lower than corresponding cloacal and subcutaneous temperatures, weekly fluctuations were lower. The weekly TC head temperatures were only slightly lower than the corresponding TS temperatures except during weeks 17–19, when differences in actual and target chamber temperatures in both regimes were small (Figure 2). Average weekly torso (surface) temperatures were the lowest of the four temperature measurements (Figure 6). The spike in week 16 TC torso and head surface temperatures was due to higher chamber temperatures (Figure 2).

Generally, as the birds aged, their head and average torso temperatures decreased (Figure 6), like broilers [22]. While airflow rate did not affect any type of bird temperature, the turkeys experienced greater heat stress when temperatures were elevated by 11 °C even for 2 h (Figure 6). Since evaporative cooling is only provided when house temperatures exceed the upper 20s (in Celsius), wind chill created by high airspeeds would be the only mechanism for cooling large tom turkeys under moderate temperature conditions (<25 °C). Hence, compared with tunnel-ventilated houses, large tom turkeys in sidewall-ventilated houses could experience greater heat stress even when ventilation rates are similar due to lower airspeed.

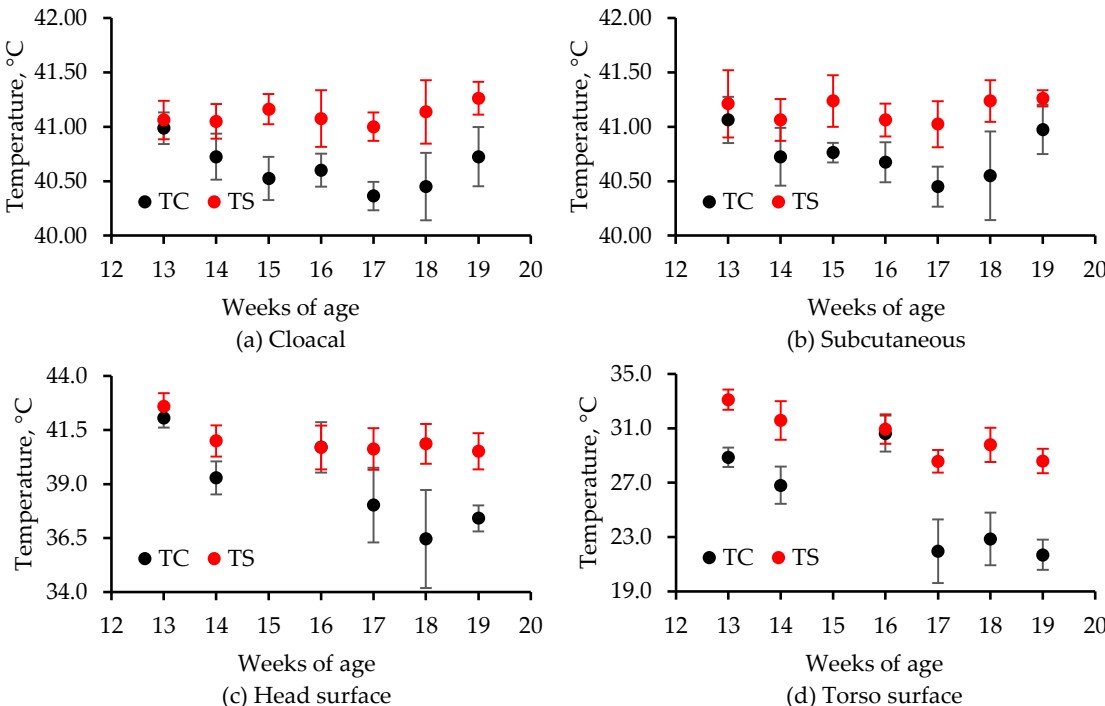

**Figure 6.** Plots of the mean (**a**) cloacal, (**b**) subcutaneous, (**c**) average head surface, and (**d**) average torso temperatures over the 7-week study after 2 h of exposure. The temperatures for each week were obtained by averaging the final values for the four airflow rates and two replicates (*n* = 8), except for (**c,d**) for weeks 16 and 19 of age for TC when *n* = 6. Infrared temperatures (**c,d**) were lost for week 15 of age. The error bars represent ± one standard deviation.

*3.2. Room Study*

In this 6 d study, the impact of the ventilation rate (Table 3) on bird performance, blood parameters, and temperatures (surface and subcutaneous) was evaluated on 20-week tom turkeys. Additional details are in Uemura [18].

3.2.1. Environmental Conditions

Summary hourly temperature and RH in the four rooms (with different ventilation rates, Table 3) are presented in Table 5, while their trends are compared in Figure 7. Temperature and RH data for the first 2 d were discarded as the birds knocked over the sensors

to the ground. Lower average temperatures (Table 5) and bigger temperature fluctuations (Figure 7) at the higher ventilation rates were due to the inability of the brooders to maintain the target temperature of 20.5 °C at VR-100. Except for VR-100, RH decreased with an increase in ventilation rate (Table 5) due to moisture removal in the ventilation air. The highest RH in VR-100 (Table 5) might have been due to higher propane combustion in that room, resulting in higher moisture addition.

**Table 5.** Summary temperature and RH data in the four treatments (rooms).

| Room | Temperature, °C | RH, % |
|------|-----------------|-------|
| VR-100 | 18.5 ± 1.7 [1] | 55 ± 4 [1] |
| VR-75 | 20.8 ± 1.1 | 38 ± 4 |
| VR-50 | 21.9 ± 0.3 | 45 ± 6 |
| Control | 22.6 ± 0.5 | 49 ± 5 |

[1] Mean ± SD for all treatments.

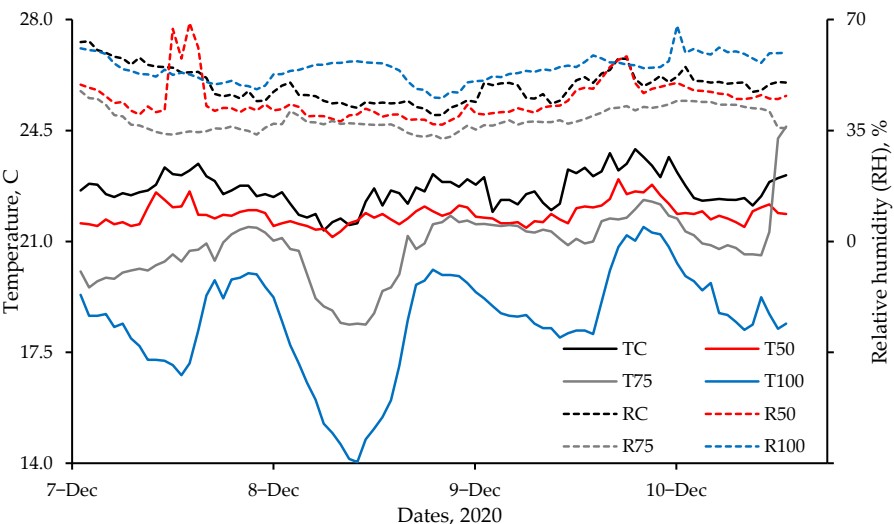

**Figure 7.** Hourly average ($n$ = 12) temperature and relative humidity (RH) trends in the four treatments, beginning the afternoon of 7 December 2020. Lines (instead of markers) are used to demonstrate the trends. In the legend, the first alphabet T (or R) represents temperature (or RH). The remainder of the legend represents the ventilation rate as: C—Control, 50—VR−50, 75—VR−75, and 100—VR−100. Earlier data were discarded. The target room temperature was 20.5 °C.

### 3.2.2. Bird Subcutaneous Temperatures

Unlike surface temperature for the whole body, head, and legs (Table 6), there was no treatment effect on subcutaneous temperature (Table 6) over the 6 d study. Two of three birds implanted with Anipill temperature sensors in the Control treatment had been in the previous high heat treatment, whereas all the birds in the other treatments had received the high heat treatment.

**Table 6.** Ventilation rate effects on average hourly subcutaneous temperatures of 20-week tom turkeys over the 6 d study.

| Treatment | Temperature, °C |
|-----------|-----------------|
| VR-100 | 39.9 ± 0.2 [1] |
| VR-75 | 40.2 ± 0.2 |
| VR-50 | 40.2 ± 0.2 |
| Control | 40.1 ± 0.2 |
| ANOVA *p* value | 0.30 |

[1] Mean ± SD ($n$ = 3).

In this study, with low to moderate room temperatures, the birds might have been able to maintain core body (hence, subcutaneous) temperatures. Subcutaneous temperatures in this study were lower than chamber subcutaneous temperatures (Figure 6), probably because these birds were older and had developed coping mechanisms to handle heat stress. When 23-week tom turkeys acclimated at 19 °C, and 65% RH were subjected to 32 °C and 65% RH for 3 weeks, their cloacal temperatures significantly increased (by 0.4 °C) at the end of week 1 but did not increase thereafter [23]. In this study, neither the room temperature nor the airspeed sufficiently differed among treatments to significantly affect subcutaneous temperatures, though there were significant impacts on surface temperatures (Section 3.2.3). Average hourly subcutaneous temperatures for the four treatments are shown in Figure 8; the average values are based on the average hourly temperatures of three birds in each treatment. Hourly temperature for each bird was obtained by averaging four 15-min readings.

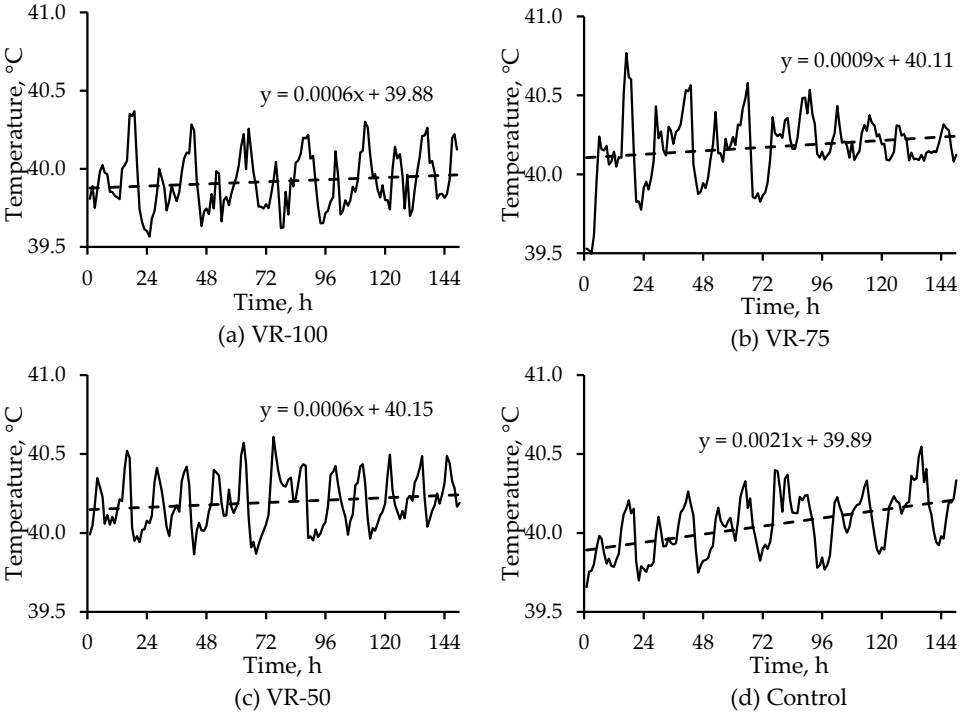

**Figure 8.** Average (*n* = 3) hourly subcutaneous temperature trends in 20-week tom turkeys in the four treatments ((**a**) VR-100, (**b**) VR-75, (**c**) VR-50, and (**d**) Control) in the 6 d room study. Lines (instead of markers) are used to demonstrate the trends more clearly. Temperatures were measured every 15 min and averaged every hour. The dashed line represents the linear fit.

Over the 6 d study, in all four treatments, the subcutaneous temperature trend lines had positive slopes, but the slope of the Control treatment was the highest. A steeper positive slope might be indicative of the reduced ability to cope with heat stress over time vs. other treatments. Generally, subcutaneous temperature peaked around mid- to late-afternoon and decreased to a minimum around midnight and then started to increase after daybreak. This was likely due to both bird activity as well as diurnal temperature variations; however, due to the absence of treatment effects, further analyses were not performed.

### 3.2.3. Bird Surface Temperatures

Average whole body, head, and leg surface temperatures were significantly affected by the ventilation rate treatment but not the torso surface temperatures (Table 7). Birds in the Control treatment had significantly higher body surface temperatures than birds in the other treatments (Table 7). Despite a 30% higher ventilation rate in VR-100, VR-75 had a significantly lower head temperature, while average body and leg surface temperatures

were not significantly different (Table 7). While VR-100 had nearly double the ventilation rate of VR-50, they were not significantly different in average body, head, and leg surface temperatures (Table 7). The VR-50 and Control treatments were not significantly different in head and leg temperatures, though the VR-50 had significantly lower average body temperature (Table 7). A higher number of replicates vs. the chamber study (6 vs. 2) might have also led to significant differences in surface temperature measurements with the IR camera between the treatments.

**Table 7.** Ventilation rate effects on 20-week tom turkey surface temperatures over the 6 d study.

| Treatment | Surface Temperature, °C | | | |
| --- | --- | --- | --- | --- |
| | **Average Body** | **Head** | **Torso** | **Leg** |
| VR-100 | 29.7 ± 1.3b [1,2] | 38.6 ± 0.4b | 28.1 ± 1.8 | 36.4 ± 1.1b |
| VR-75 | 29.0 ± 1.0b | 37.7 ± 0.7c | 27.6 ± 0.9 | 35.9 ± 0.8b |
| VR-50 | 29.9 ± 1.1b | 39.0 ± 0.7b | 28.1 ± 1.9 | 37.4 ± 1.2ab |
| Control | 31.6 ± 0.9a | 39.8 ± 0.6a | 29.4 ± 1.1 | 38.2 ± 0.9a |
| ANOVA *p* value | <0.01 | <0.001 | 0.15 | <0.01 |

[1] Mean ± SD (*n* – 6). [2] Different alphabets (a–c) in the column indicate that the treatments are significantly different using the Tukey's HSD at α = 0.1.

While convective heat loss (and hence, reduction in surface temperature) increases with airspeed [22], that was not always observed in this study (Table 7), which could be due to two reasons. First, despite having adequate ventilation rates, all treatments had airspeeds below still air threshold airspeed of 0.25 m·s$^{-1}$ [24]. Even though the plastic curtain opening increased airspeed by reducing cross-sectional area (vs. the room cross-section) (Figure 3), it was still inadequate. Finally, the difference in air temperature and RH values in the four treatments (Table 5) could have masked the main treatment effect (ventilation rate). For example, despite being 2.3 °C cooler, head surface temperature was significantly higher in VR-100 vs. VR-75; it is unclear if this was due to 17% lower RH in VR-75 (Table 5), which could have increased latent heat loss.

The smallest significant difference in surface temperature and, hence, heat stress change measured by the IR camera was 0.8 °C, between the VR-50 and Control head surface temperatures (Table 7). This was due to very small differences in room air temperature and RH between the treatments (Table 5). If the VR-50 and Control rooms had the same temperature, and RH and the Control room had zero airspeed, subtracting the head surface temperature in VR-50 from the corresponding Control value would have yielded (at least, the short-term) wind chill effect due to VR-50. The 0.8 °C temperature difference (between VR-50 and Control) is the same as a wind chill of 0.8 °C at an airspeed of 0.8 m·s$^{-1}$ vs. still air on the USDA wind chill curve for 7-week broilers in an airspeed range of 0 to 1.02 m· s$^{-1}$ [25]. Since the slope (°C per m·s$^{-1}$) of the wind chill curve increases with airspeed, at higher airspeeds, the wind chill resolution as measured with the IR camera would improve. This shows that the low-cost camera can be used to develop poultry wind chill curves. A wind chill is specific to the age (size) of a particular poultry bird. There are no wind chill curves for turkeys or for other ages of broilers. Mature poultry are greatly affected by heat stress, e.g., in July 2012, in a 10-county region of eastern North Carolina, 342,000 mature turkeys died due to heat stress (James Parsons, personal communication, 20 August 2012). The use of IR thermography to develop wind chill curves for different ages and types of birds would be relatively simple and useful.

An IR camera with a higher resolution than the FLIR E8 might improve wind chill resolution. A machine vision system with an IR camera could collect the head surface temperatures of several birds at once and, after adjusting for airspeed, air temperature, RH, and bird age, could calculate heat stress or wind chill. The system could also instruct the house environmental controller to adjust the ventilation rate and/or evaporative cooling in real time. Such optimization might improve bird comfort and performance and reduce electricity use. However, considerable work would be required prior to deployment. For

example, an age-based wind chill curve would be required. As is clear from Table 7, the head was the warmest part of the turkey's surface. To reduce uncertainty in image delineation, averaging the maximum head temperatures (single pixel) should be considered. Once developed, the model would require training, testing, and validation prior to deployment. It might be possible to have an IR camera as part of a comprehensive monitoring system similar to Big Dutchman's ChickenBoy® [26], mounted on rails and suspended from the ceiling. At this time, the ChickenBoy® does not have such capability. With suitable training, such an IR camera-enabled machine vision system might also detect outbreaks of some diseases that change the body (hence, surface) temperature, e.g., coccidiosis [27].

3.2.4. Bird Performance and Hematology

There was a significant treatment effect ($\alpha$ = 0.1) on bird weight gain (Table 8), but only VR-100 was significantly greater than VR-50. As mentioned in Section 2.2.2, the initial bird weights were determined 2 d prior to the start of the study, and therefore, the weight gains in Table 8 are for 8 d. However, the comparison of weight gains among the treatments was considered valid since for the first 2 d, the birds received the same ventilation treatment.

**Table 8.** Treatment effect on bird weight gain and hematology.

| Treatment | Weight Gain [1,2], kg | Hematocrit [1], % | Corticosterone [1], $pg^{-1} \cdot mL^{-1}$ |
|---|---|---|---|
| VR-100 | 2.34 $\pm$ 0.28a [3] | 0.5 $\pm$ 0.0a | 3327 $\pm$ 1866a |
| VR-75 | 2.14 $\pm$ 0.43ab | 0.5 $\pm$ 0.0 | 3387 $\pm$ 887 |
| VR-50 | 1.79 $\pm$ 0.39b | 0.6 $\pm$ 0.0 | 3703 $\pm$ 1400 |
| Control | 1.93 $\pm$ 0.21ab | 0.5 $\pm$ 0.0 | 2857 $\pm$ 1079 |
| ANOVA *p* value | 0.08 | 0.21 | 0.78 |

[1] *n* = 6. [2] For 8 d. [3] Different alphabets (a,b) in the column indicate that the treatments are significantly different using Tukey's HSD at $\alpha$ = 0.1.

Significantly higher weight gain only in VR-100 vs. VR-50 was unexpected. The lack of treatment effect (weight gain) among the other treatments may have been due to the small difference in airspeeds between the treatments as well as the short duration of the study (Table 8). As mentioned in Section 2.2.3, due to missing feed data, FCR could not be calculated. While the room temperatures were much higher than the comfortable temperature of large turkeys recommended by Aviagen [28], in terms of weight gain, the turkeys performed much better. For reference, 21-week Nicholas tom turkeys have an average daily weight gain (ADWG) of 163 g over a 1-week period [29], whereas ADWG in this study varied from 224 g (VR-50) to 293 g. Most of the birds used in this study came from a heat stress study, and cooler temperatures (particularly in VR-100) in this study may have led to increased feed intake and, hence, ADWG in this study.

There were no statistically significant differences among the treatments with respect to hematocrit (ANOVA *p* = 0.16) or corticosterone (ANOVA *p* = 0.22) concentrations (not presented) at the start of the study. There was no treatment effect on blood hematocrit and corticosterone levels at the end of the study (Table 8). While hematocrit decreases [30] and corticosterone increases [31] in response to heat stress, differences in airspeeds among the treatments were likely too small to cause hematological changes. Corticosterone concentrations measured at the end of this study were lower than values (~5000 $pg^{-1} \cdot mL^{-1}$) reported in the literature for tom turkeys of similar age [32].

## 4. Conclusions

Chamber and room studies were used to assess heat stress at moderate temperatures (<25 °C) and low airspeeds on grown tom turkeys. Subcutaneous, cloacal, and IR temperatures were used, whereas blood chemistry and weight gain were also used in the room study to assess heat stress. In the chamber study, a combination of four ventilation rates and two temperatures (thermal comfort and heat stress) were applied to 13- to 19-week

birds. Despite large differences in chamber airflow rates, the lack of treatment effect on any temperature measurement was due to low airspeeds, which were similar among the treatments. Subcutaneous temperature was a better quantitative indicator of short-term (2-h) heat stress than cloacal temperature. In a 6-day room study, where the target room temperature was 11 °C above the comfortable temperature of 21-week toms, four ventilation rates were used. The VR-100, VR-75, and VR-50 treatments had significantly lower whole-body temperatures (IR) than the Control treatment, though VR-100, VR-75, and VR-50 treatments did not significantly differ from one another. With respect to the head IR temperatures, it was VR-75 > VR-100 $\approx$ VR-50 > Control treatment; there was no treatment effect on subcutaneous temperature. Weight gain was significantly higher in the VR-100 vs. VR-50, whereas there were no significant differences between the other pairs of treatments. There was no treatment effect on hematology. Unlike the chamber study, in the room study, due to the larger number of replicates, the low-cost IR camera effectively detected surface temperature differences as low as 0.8 °C, resulting from small temperature and RH differences but not airspeed differences that were very low. Hence, the FLIR E8 camera could measure small differences in heat stress. Based on the USDA wind chill curve for 7-week broilers, the IR camera could measure wind chill as low as 0.8 °C at an airspeed of 0.8 m·s$^{-1}$ vs. still air. The combination of machine vision with IR thermography could allow for measuring and mitigating poultry heat stress in real time. Such a system might also be useful in disease detection.

**Author Contributions:** Conceptualization, S.S., D.U. and J.G.; methodology, D.U., S.S., P.R. and J.G.; formal analysis, D.U., P.R. and S.S.; investigation, D.U. and S.S.; resources, S.S., P.R. and J.G.; data curation, D.U.; writing—original draft preparation, D.U.; writing—review and editing, S.S., P.R., L.W.-L. and J.G.; visualization, S.S.; supervision, S.S. and J.G.; project administration, S.S.; funding acquisition, S.S. All authors have read and agreed to the published version of the manuscript.

**Funding:** This research was funded by the North Carolina Ag Foundation, grant numbers 21-25 and 20-17.

**Institutional Review Board Statement:** The turkeys used in the study were managed according to the experimental protocols (IACUC # 18-155-A) approved by the University's Institutional Animal Care and Use Committee.

**Informed Consent Statement:** Not applicable.

**Data Availability Statement:** Consult D. Uemura's thesis, listed in the references. Raw data are available upon request from the corresponding author, S. Shah.

**Acknowledgments:** The North Carolina Ag Foundation provided funding for this project. Supplemental support to D.U. was provided by North Carolina State University's Biological and Agricultural Engineering (BAE) Department. The BAE Department's Research Shop fabricated the chambers. Support provided by Stephen Hocutt and the entire staff and students working at North Carolina State University's Talley Turkey Education Unit is gratefully acknowledged. Prestage Farms provided the turkeys for the chamber study. Technical advice provided by Isaac Singletary of Munters Aerotech is gratefully acknowledged. The authors thank MDPI for waiving the OPC.

**Conflicts of Interest:** The authors declare no conflict of interest. The funders had no role in the design of the study; in the collection, analyses, or interpretation of data; in the writing of the manuscript; or in the decision to publish the results.

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
