# Peer review of "Low Airspeed Impacts on Tom Turkey Response to Moderate Heat Stress"

_agriengineering, doi:10.3390/agriengineering5040121_

Round 1
Reviewer 1 Report
Comments and Suggestions for Authors
The article “Low Airspeed Impacts on Tom Turkey Response to Heat Stress” by Uemura et al., is well designed and written study. Min be or comments to addressed
1. Lines 19-20: the conclusion is irrelative to the title of the paper .please add a conclusion of the main study
2. Animal grouping in table, why the control group is zero air flow rates, shouldn’t it be the normal air flow rate recommended for turkey please justify
3. What is the difference between table 1 and table 3 I see here an airflow rate for control group
4. Discussion: Please discuss the study findings related to effect of heat stress on intestinal integrity that consequently affect fcr in the light of (10.51585/gjvr.2023.1.0051 and 10.3390/nu12030734 and 10.51585/gjvr.2022.3.0040
5. For the conclusion related to IR camera, is this objective was planned, if so please mention in the study objectives, and I believe I may require an indication in the title of the study as a very important noninvasive mean for temp. Monitoring
Reviewer 2 Report
Comments and Suggestions for Authors
The authors present data from a study titled „Low Airspeed Impacts on Tom Turkey Response to Heat Stress“. The manuscript is written in great detail, with clearly stated objectives and methodology. The results are presented in detail, and the discussion is scientifically informed. The conclusions are clearly drawn.
Specific comments and suggestions:
L 24 – L 112 The Introduction section is too long. It should be shortened. Some of the sentences can be moved to the Discussion section or removed.
L 125 – L 134 A picture or drawing of the chamber where the turkeys were kept during the experiment should be added after this paragraph.
Reviewer 3 Report
Comments and Suggestions for Authors
The second half of the abstract can be written more clearly. The introduction may contain more concise information and justify using moderate temperatures for the experiment. The conclusion of a scientific article is a critical section where you summarize the key findings and insights from your research. It should provide a clear and concise summary of what you have discovered and the implications of your findings.
It is pertinent to place a chapter with statistical analysis that justifies the use of parametric or nonparametric statistical tests (including, for example, a discussion of data normality when the test is appropriate only for normal data), alpha level for all tests if tests were one-tailed or two tails. The text of the results should indicate the actual P value for each test (not simply "significant" or "alpha value 0.1"). Clarify which statistical test was used to generate each P value. In the experiment, an alpha value of 0.1 was used; in physiology, alpha values of 0.05 or 0.01 are used; the use of 0.1 must be strongly justified.
Establishing what environmental temperature is considered heat stress is important since the text mentions moderate temperatures, and the title indicates heat stress.
The experiment performed superficial body measurements and not systemic ones. In order to evaluate the response to heat stress, it is necessary to quantify body weight gain and obtain radio feed conversion.
For the research to be published, the title and text of the scientific article must be modified to match the experimental design.
Comments on the Quality of English LanguageThe quality of the English language is sufficient to publish the research; however, comments from this revision should be considered for the new text.
Round 2
Reviewer 3 Report
Comments and Suggestions for Authors
The right work for publication. It might have more field variables to consider, but it's already ready for publication.